# Mental Health Correlates of Autistic and ADHD Traits in Secondary School Students

**DOI:** 10.3390/brainsci15060609

**Published:** 2025-06-04

**Authors:** Japnoor Garcha, Andrew P. Smith, Arwel James

**Affiliations:** 1Department of Occupational and Health Psychology, Cardiff University, 70 Park Place, Cardiff CF10 3AT, UK; 2School of Psychology, Cardiff University, 70 Park Place, Cardiff CF10 3AT, UK; smithap@cardiff.ac.uk (A.P.S.); arweljames@rhydywaun.org (A.J.)

**Keywords:** ADHD traits, anxiety, autistic traits, depression, secondary school students

## Abstract

**Background:** Recent research has examined the associations between autistic traits and the well-being of students. This research has also included measures of ADHD and mental health. **Methods:** To understand the association and interaction of well-being and mental health with autistic traits (AQ) and attention deficit hyperactivity disorder (ADHD) traits, a survey was given to 578 secondary school students. The survey used the well-being process questionnaire (WPQ), the autism spectrum quotient, the ADHD self-report scale, and the Strengths and Difficulties questionnaire (SDQ). **Results:** The analysis conducted using SPSS showed that there was a significant correlation between anxiety, depression, AQ, and ADHD. Anxiety and depression were also significantly correlated with all well-being and SDQ variables. The regression analysis, including psychosocial predictors of well-being, anxiety, depression, ADHD, and AQ, showed that anxiety and depression were strongly associated with well-being outcomes. In contrast, AQ and ADHD were associated with hyperactivity, peer problems, and reduced prosocial behaviour. The associations between anxiety, depression, and well-being outcomes were stronger than with either AQ or ADHD. **Conclusions:** This study extends previous research from university samples to a secondary school sample. This shows the importance of investigating associations between subclinical traits and psychological symptoms in early adolescence, as this will lead to better-informed prevention and early intervention strategies.

## 1. Introduction

Autism spectrum disorders have been classified as neurodevelopmental disorders and have seen an increase in prevalence in recent years. It has increased from 1 in 5000 in the 1970s to 1 in 36 in 2023 [1]. Autism spectrum disorders are identified and evaluated over specific characteristics, with the main categories being social impairments and repetitive behaviour as per DSM-5 [2]. Population studies [3] have found that many individuals exhibit subthreshold or autistic-like traits, characterised by social communication difficulties and repetitive behaviours. Still, these are not significant enough for a formal diagnosis [4]. This has strengthened the research belief that autistic traits exist throughout the population. This has also led to studies linking autistic characteristics with risk factors for mental health and mental health symptoms. The literature quoted here focuses on the presence of autistic traits in the population and the prevalence and association of autistic traits with other mental health problems. A longitudinal study found that children aged 8, 10, and 13 years who scored higher on autistic traits had a greater prevalence of bullying. The study also found that children aged ten who had autistic traits also scored higher than average on depressive symptoms [5]. Other studies [6,7] have found a correlation between autistic traits and psychotic traits. The presence of autistic-like traits in people diagnosed with MDD and schizophrenia has also been observed [8]. Another study also found an association of autism-like traits with conduct problems, ADHD, anxiety, and depression [9]. Research has shown that older adults with higher/elevated autistic traits are more prone to poorer mental health, especially later in life [10]. A study also found that patients with borderline personality disorder had higher autistic traits [11]. In a survey conducted on U.K. students, high levels of autistic traits were associated with increased social anxiety [12]. There has also been sufficient research that establishes a link between autism and other disorders, such as ADHD. However, research focusing on the co-occurrence of autistic and ADHD traits in the general population, rather than diagnosed cases, is a topic requiring further focus and investigation. 

The WHO has said that mental health disorders are now the leading cause of disability worldwide. Instead of updates and inclusivity, an all-inclusive definition of mental health was devised: “Mental health is a dynamic state of internal equilibrium which enables individuals to use their abilities in harmony with universal values of society. Basic cognitive and social skills, ability to recognise, express and modulate one’s own emotions, as well as empathise with others, flexibility and ability to cope with adverse life events and function in social roles, and a harmonious relationship between body and mind represent important components of mental health which contribute, to varying degrees, to the state of internal equilibrium.” [13]. The World Health Organization has also formulated a new definition of well-being: “a state of well-being in which the individual realises his or her abilities, can cope with the normal stresses of life, can work productively and fruitfully, and can contribute to his or her community” [14]. Emotional, social, and psychological well-being are three components of well-being. 

Research has shown that autistic traits are significant predictors of well-being; however, there are other links and predictors of well-being, especially in the context of autism [15].

The present study is focused on autistic traits present in secondary school students. Hence, it has looked at the presence of autistic traits using self-report survey scales and not diagnosed cases of autism spectrum disorders. Hence, the results mentioned are not representative of people with a diagnosis of autism. The following section is a summary background review to understand this research on autistic traits without an autism diagnosis and serves as a strong precursor to understanding current research.

## 2. Rationale and Context for the Literature

To investigate the complex relationships between autistic traits and student well-being, this research draws upon a theoretical framework grounded in the well-being process model (WPM) [16]. The WPM, adapted from the DRIVE model [17] originally used in occupational settings, adopts a holistic perspective by examining both positive (e.g., life satisfaction, happiness) and negative outcomes (e.g., stress, anxiety, depression). It provides a flexible structure that allows for the inclusion of individual differences, coping styles, and contextual variables, making it suitable for student populations.

There is an increasing body of literature highlighting the relationship between autistic traits and mental health outcomes, such as anxiety and depression [18,19]. Studies in both clinical and non-clinical populations have shown that higher levels of autistic traits—measured by tools like the Autism Quotient (AQ)—are associated with lower subjective well-being and life satisfaction [20]. However, these studies often lack consideration of broader psychosocial predictors or focus only on diagnosed populations, overlooking those with subclinical traits in general student populations.

This study addresses that gap by using the AQ scale to dimensionally explore autistic traits, based on the premise that these traits exist along a spectrum in the general population [21]. Previous work conducted by the author demonstrated associations between autistic and ADHD traits and mental health outcomes among university and secondary school students [22]. These initial studies revealed that while both traits were associated with well-being, the inclusion of established predictors such as stress and coping styles modified those associations.

Further analysis conducted in this study mentioned above explored whether the combined effects of autistic traits, ADHD, and mental health issues like anxiety and depression were stronger than their individual effects [23]. A composite variable representing this combination was found to be the most robust predictor of negative well-being and behavioural difficulties (e.g., hyperactivity, conduct problems), suggesting that comorbidity may significantly compound mental health challenges. The present study moved the focus from university students to secondary school students, as adolescence is a crucial time of development. This is also the stage when students experience an increase in academic pressure, challenges with identity and constantly changing peer dynamics, and all this can significantly impact their well-being and mental health. This can also be linked to studies that show more pronounced mental health problems in adolescence. This makes secondary school students a meaningful and important population to study. Hence, the focus of the current study on secondary school students who are at a more malleable developmental stage builds on prior research and can lead to the establishment of early interventions which may be most impactful.

In summary, this research integrates a robust theoretical model (WPM), employs validated psychometric instruments (e.g., AQ, ASRS, WPQ, SDQ), and responds to a clear gap in the literature by focusing on students with varying levels of autistic traits. It extends prior research by simultaneously considering established and emerging predictors of well-being, offering a nuanced and comprehensive approach to understanding mental health in educational settings.

The above studies lead to the following hypotheses:The first was that autistic and ADHD traits would be significantly positively correlated.The second was that the associations between well-being predictors, well-being outcomes, strengths and difficulties, AQ, ADHD, and mental health traits would be significant, with higher levels of autistic and ADHD traits linked with higher levels of anxiety and depression and lower well-being.The next was that on controlling for established correlates of well-being, the associations between autism, ADHD, anxiety, and depression traits and well-being and strengths and difficulties would remain consistent and significant.The final was that when established predictors are covaried, some outcomes will remain associated with AQ and ADHD.

Based on the literature reviewed above and the research gaps identified, the current study sets out the following aims:To examine the associations between autistic and ADHD traits and mental health outcomes of anxiety and depression in secondary school students.To investigate the individual and combined effects of autistic traits, ADHD traits, and mental health symptoms on well-being outcomes.To assess whether the associations between traits and outcomes remain constant after controlling for established predictors of well-being (stress, social support, workload, work–life balance, positive coping, negative coping, psychological capital, flow, rumination).

## 3. Materials and Methods

### 3.1. Consent

This research was carried out with the informed consent of the participants and the approval of the Ethics Committee, School of Psychology, Cardiff University.

### 3.2. Participants and Sample Size

The participants were all students studying in a secondary school in South Wales. Five hundred and fifty-six students took part in this study. There were 276 males and 280 females, with ages ranging from 11 years to 18 years. The year groups they belonged to are as follows:Year 5 = 1;Year 6 = 90;Year 7 = 124;Year 8 = 87;Year 9 = 96;Year 10 = 97;Year 11 = 27;Year 12 = 20;Year 13 = 3.

Thirteen students did not fill out their year in the survey. Twenty-two participants did not fill out the entire survey, leaving some of it blank, which makes *N* = 578.

### 3.3. Questionnaires

AQ-10

Simon Baron-Cohen designed the autism spectrum quotient [21] using a self-report scale. The original questionnaire was a 50-item questionnaire; over time, shorter versions have been created, and the 10-item version was used. It measures the autistic traits in an individual. The scores range from 0 to 10 and are based on a 4-point Likert scale ranging between ‘definitely agree’ and ‘definitely disagree’. Total scores were used as a measure of AQ. The AQ is scored dichotomously (0 or 1); hence, when the total score is calculated, it ranges from 0 to 10. The AQ 10 has a Cronbach alpha of 0.7.

2.ADHQ

The ADHQ [24] is a self-report scale known as the adult ADHD self-report scale (ASRS), which was designed and formulated in collaboration with the WHO. It consists of 18 questions and uses a five-point rating scale from never to very often. It has been constantly used as a diagnostic measure. The scale is divided into two parts, with the first having six questions most predictive of ADHD. Studies have been conducted to see if ASRS is well-suited for use with adolescents. A study conducted in 2018 found the ASRS to be reliable for use with middle and high school students [25]. Research on ASRS has also suggested that it is a reliable tool to collect information regarding the presence of current symptoms of ADHD in college and university students [26]. The ASRS was also found to be psychometrically sound and reliable for use in adolescents, even in an Italian translated version [27]. The ASRS is divided into two parts, Part A and Part B. Part A consists of 6 items that screen for the core symptoms of ADHD. The total score for the entire scale and the individual total score for part A are calculated, and the part A score is used in analysis as it is directly reflective of core ADHD symptoms. This scale has a Cronback alpha of 0.8.

3.SDQ

The Strengths and Difficulties Questionnaire was designed by Goodman [28]. It comprised 25 items spread over five subscales: conduct problems, emotional symptoms, hyperactivity/inattention, peer problems, and prosocial behaviour. A short form with single items for each subscale was used here. Responses were made on a 10-point scale. The short form of the Strengths and Difficulties Questionnaire is shown in Appendix A. The short form of the SDQ has been used, as single items are significantly correlated with the full scales, with the total SDQ score having a Cronbach alpha of >0.7.

4.Short form Student WPQ

The short-form student WPQ [29] was adapted from the well-being process questionnaire. The short form consisted of 14 items and also had single-item questions on anxiety and depression. The questionnaire contains predictor variables: student stressors, negative coping, workload, work–life balance, daytime sleepiness, psychological capital, social support, positive coping, negative coping, and flow. The dependent variables were positive well-being, negative well-being, physical health, and the extent to which the person was flourishing. The WPQ measures anxiety and depression with single questions that have been shown to be highly correlated with the HAD scales [30]. Positive well-being outcomes have a Cronbach alpha of 0.79, negative outcomes a Cronbach alpha of 0.88, positive well-being predictors a Cronbach alpha of 0.73, and negative well-being predictors a Cronbach alpha of 0.70.

It is important to make note before moving forward that well-being refers to well-being, whereas mental health refers to anxiety and depression. SDQ variables are used to understand specific types of difficulties like hyperactivity, emotional problems, peer problems, prosocial behaviour, and conduct problems.

### 3.4. Procedure

An online survey with a cross-sectional design was conducted using the Qualtrics platform. The students were all asked to complete the survey online, and the presence of autistic and ADHD traits was assessed using the AQ and ASRS questionnaires. The survey was self-paced and took about 20–25 min to complete. The order was standardised for all participants. The data were collected between October 2022 and November 2022.

### 3.5. Analysis

This study used different analysis strategies using the SPSS software version 27. The various analyses helped in understanding in detail the associations between all the variables. The first analysis was the descriptive analysis that was applied to all variables. Then, Pearson’s correlation analysis was performed to see the associations between autism and ADHD. A set of linear regressions was also carried out where the first set had AQ and ADHD as predictor variables. This was followed by the second set, which had predictors from the well-being process questionnaire. The third set had AQ, ADHD, and established predictors. The final set of regression analyses was to see the association between AQ, ADHD, well-being, and mental health and the associations between well-being, strengths and difficulties, and mental health.

## 4. Results

The results for the analysis are listed below. Multiple testing should be adjusted for only where authors use the significance of statistical tests to weight the reporting, discussion, and interpretation of their findings. We have not performed this.


**Descriptive Statistics.**


Table 1 shows the descriptive statistics for both outcomes and predictor variables. There was considerable variation between each of the measures. Very little data were missing, and all scores were comparable to previous findings. Six participants left a significant part of the questionnaire blank and were not included in the analysis. For all the analyses, only the cases were considered where all questions had been answered. This led to a variability in *N* for different analyses.

2.
**Associations Between AQ, ADHD, Mental Health, and Outcome Variables.**


The correlations between AQ and ADHD are shown in Table 2. AQ and ADHD were significantly correlated with all outcome variables. The outcome variables included variables from the WPQ, namely positive and negative well-being, physical health, flourishing, and the SDQ outcomes of conduct, hyperactivity, emotional, peer, and prosocial behaviour.

3.
**Association Between AQ, ADHD, Mental Health, and Predictors of Well-Being.**


Table 3 shows the correlations with the established predictors of well-being, variables on the WPQ that have repeatedly been associated with the outcomes in the previous literature. Each independent variable was correlated against a dependent variable to see the correlation and internal associations. AQ was significantly correlated with all variables except work–life balance and rumination. ADHD was significantly correlated with all variables. AQ and ADHD were also significantly correlated. Stress was significantly correlated with all variables. Social support was significantly correlated with all variables except work–life balance. Positive coping was significantly correlated with all variables except work–life balance. Negative coping was significantly correlated with all variables. Psychological capital was significantly correlated with all variables. Work–life balance was significantly correlated with all variables except AQ, social support, positive coping, flow, and rumination. Workload was significantly correlated with all variables. Sleepiness was significantly correlated with all variables except rumination. Flow was significantly correlated with all variables except work–life balance. Rumination was significantly correlated with all variables except AQ, work–life balance, and sleepiness.


**Regression Analysis with AQ and Established Predictors.**


The regression analysis with autistic traits (AQ) and established predictors of well-being was run to see which established predictors remove the effects of AQ. It was seen that AQ had no significant associations with any of the established predictors. Multicollinearity diagnoses were performed, and multicollinearity was assessed using the variance inflation factor (VIF). All variables had a VIF less than 10. As VIF < 10, no multicollinearity was found. The VIF scores are added in the regression tables below.

4.
**Regression with AQ, ADHD, Anxiety, and Depression as Predictor Variables.**


The regression analysis aimed to examine associations between variables. The results for regression are shown in Table 4 below. Each outcome and the established predictor had separate regression analyses. Positive well-being and negative well-being were predicted by anxiety and depression. Anxiety also predicted emotional problems and prosocial behaviour. Depression predicted physical health, flourishing, conduct problems, emotional problems and peer problems. Autism only predicted hyperactivity, and ADHD predicted hyperactivity and conduct problems.

It was essential to see the associations between AQ and ADHD when the established predictors were covaried and compare this association with previous studies conducted on secondary school and university students.

5.
**Regression with AQ, ADHD, and Established WPQ Predictors.**


Separate regressions were carried out for each outcome, which are shown in Table 5 below. The established predictors of WPQ have been generated based on previous research studies that used WPQ. AQ predicted the outcome only as positive coping. ADHD, which was another independent variable, predicted stress, social support, positive coping, work–life balance, workload, sleepiness, and flow. It was seen in the previous studies with university students that AQ and ADHD were associated with three SDQ outcomes, namely hyperactivity, peer problems, and conduct problems when established predictors were covaried. The secondary school study associated AQ with conduct problems and prosocial behaviour. ADHD scores were associated with hyperactivity.

6.
**Regression with AQ, ADHD, Anxiety, depression, and Established WPQ Predictors.**


Separate regressions were carried out for each outcome, and are shown in Table 6 below. Positive well-being was predicted by depression, social support, and psychological capital. Negative well-being was predicted by depression, stress, psychological capital, and sleepiness. Physical health was predicted by psychological capital. Flourishing was predicted by depression, psychological capital, flow, and rumination. AQ, ADHD, and workload predicted hyperactivity. ADHD predicted conduct problems. Emotional problems were predicted by anxiety, depression, stress, negative coping, and sleepiness. Peer problems were predicted by stress and psychological capital. Prosocial behaviour was predicted by positive coping and flow.

## 5. Discussion

The present study examined the association between autistic and ADHD traits with well-being and mental health in secondary school students. The research on students or children with autism and ADHD diagnoses is ongoing. However, not enough consideration is given to students exhibiting autistic and ADHD symptoms without a diagnosis due to the symptoms not being severe enough for a diagnosis. In today’s day and age, students are increasingly facing issues that impact them and also lead to mental health disorders. Mental health is increasing in importance every day. According to WHO, in 2019, one in eight people was living with a mental health disorder, with anxiety and depression being the most common disorders. The present study examined both the individual and combined effects of autism, ADHD traits, mental health, and well-being. In this preliminary study, no adjustment was made for multiple statistical comparisons. Small effects should, therefore, be treated with caution. This applies largely to the correlations rather than the regression analyses. The previous study with university and secondary school students showed insight into the associations between AQ and ADHD. The profile of results from the three studies is summarised below.

The three studies all showed significant correlations between autistic and ADHD traits.The AQ and ADHD traits had different results for the three studies as listed below.
(a)In the two studies, one on secondary school students and one on university students, ADHD traits were significantly correlated with all variables except prosocial behaviour.(b)AQ was significantly correlated with hyperactivity, peer problems, and prosocial behaviour in both the studies.(c)In the third study with secondary school students, AQ was significantly correlated with all variables except work–life balance and rumination.(d)ADHD traits were significantly correlated with all variables in the third study with secondary school students.In the first study with secondary school students, AQ and ADHD traits were associated with predictors of well-being and SDQ outcomes. In the study with university students, AQ and ADHD traits were associated with anxiety and depression. In the third study with secondary school students, AQ, ADHD traits, anxiety, and depression were all significantly correlated.

Variables except work–life balance and rumination. ADHD was significantly correlated with all variables. Further regression analysis with the outcome and predictor variables revealed some significant associations. AQ only had associations with hyperactivity, and ADHD only had associations with hyperactivity and conduct problems. Anxiety had substantial associations with positive well-being, negative well-being, emotional issues, and prosocial behaviour. Depression had significant associations with positive well-being, negative well-being, physical health, flourishing, conduct problems, emotional problems, and peer problems. Indeed, anxiety and depression had stronger associations with the outcomes than either AQ or ADHD. This study forms the basis for devising future research to understand further and look at the experiences of those with autistic and ADHD traits. The future research can also look at differences and compare results for those diagnosed with autism and those without to see the difference in experiences. This will lead to the creation of holistic well-being practices for students. It is also essential to see the combined effect of autism, ADHD, anxiety, and depression, as they all were significantly correlated. This also leads to a crucial question as to what the differences will be when AQ is combined with ADHD from ADHD alone or when AQ is combined with mental health from mental health alone (which includes anxiety and depression). Future research must also use moderation and mediation analysis to see if mental health is predicted. This analysis will provide a robust outlook of the results.

To further understand the results by looking at the results from previous studies, it was seen that the first study conducted on secondary school students in Wales in 2022 showed that on including established predictors in the regression analysis, there were no significant effects of AQ and ADHD on well-being outcomes. The second study conducted with university students on well-being and mental health found that anxiety and depression as mental health variables were significantly correlated with AQ, ADHD, conduct, hyperactivity, emotional, peer, positive well-being, negative well-being, physical health, and flourishing. Further analysis of AQ, ADHD, and established predictors revealed that some associations remained significant.

Based on the study with university students, the present study extended the research with secondary students. Initial correlation analysis with outcomes and predictors of well-being revealed that anxiety and depression were correlated and that they were correlated with all variables except prosocial behaviour. AQ and ADHD were significantly associated with all outcome variables. AQ was significantly correlated with all mental health problems in two nationwide twin cohorts of children and adults. The associations between neurodevelopmental traits and well-being outcomes suggest that subclinical autistic traits may impact students’ mental health. Hence, these findings can be used to inform school-based mental health interventions, especially in providing support to students exhibiting elevated autistic and ADHD traits even in the absence of a formal diagnosis. Interventions can focus on the areas found to be associated with these traits, areas such as stress management, peer relationships, and emotional regulation. This study paves the way to look at the relationship between these variables using a moderation and mediation analysis, which is the subject of another research paper.

## 6. Limitations

This research also has limitations, such as AQ being associated with conditions other than autism, which leads to overlapping traits, so there is a need to use another measure for AQ. The current study did not have any people with diagnosed autism, so to further understand this, this research can be replicated using other samples which have a clinical diagnosis. The cross-sectional design of this study also limits the ability to make causal inferences. Although associations between variables are identified, it remains unclear if the traits are preceding or actually a result of the well-being outcomes. Another limitation is that data were collected through self-reported measures, which are subject to certain biases like social desirability and recall inaccuracy. Another limitation to be considered and improved when conducting future research is the lack of multicollinearity analysis, and this was due to this study focusing on detecting significant associations and effect patterns. This study was conducted on a sample from South Wales, and thus a broader generalisation to a diverse population may be limited.

## 7. Conclusions

This study adds to the growing body of research on neurodevelopmental traits by exploring the associations between autistic traits, ADHD traits, mental health, and well-being in a secondary school population. The findings suggest that even subclinical traits of autism and ADHD are significantly linked to adverse mental health outcomes such as anxiety and depression, with mental health variables showing the strongest associations with well-being. These results underscore the importance of identifying and supporting adolescents who may not meet diagnostic thresholds but still experience considerable psychosocial difficulties. This study highlights the need for school-based screening and intervention strategies that address a broader spectrum of neurodevelopmental characteristics and emotional challenges. Future longitudinal and mediation studies are needed to understand causal pathways and inform early, tailored interventions for at-risk youth.

## Figures and Tables

**Table 1 brainsci-15-00609-t001:** Descriptive statistics (possible range 1–10 unless indicated).

Groups	Variable	*N*	Mean	Std Deviation
WPQ Variables	Positive Well-Being	578	6.52	2.191
	Negative Well-Being	577	5.08	2.597
	Stress	559	5.11	2.861
	Social Support	560	6.24	2.794
	Positive Coping	557	5.71	2.595
	Negative Coping	556	5.84	2.663
	Psychological Capital	561	6.12	2.454
	Work–Life Balance	558	4.92	2.960
	Workload	554	5.04	2.672
	Sleepiness	554	6.35	2.694
	Flow	540	5.67	2.076
	Rumination	551	4.61	2.438
SDQ Outcomes	Hyperactivity	550	6.48	2.742
	Conduct Problems	552	4.01	2.568
	Peer Problems	552	7.04	2.434
	Emotional Problems	552	5.25	2.788
	Prosocial Behaviour	553	7.54	2.096
Other variables	Anxiety	549	5.35	2.703
	Depression	552	3.66	2.742
	Total AQ	532	4.6861	1.82398
	Total ADHD	613	3.5742	1.74515
	Physical Health	552	6.13	2.087
	Flourishing	553	6.07	

**Table 2 brainsci-15-00609-t002:** Correlations between anxiety, depression, autism, and ADHD with outcome variables.

	Total AQ	Total ADHD
Total AQ	1	0.310 **
Total ADHD	0.310 **	1
Conduct	0.180 **	0.344 **
Hyperactivity	0.330 **	0.524 **
Emotional	0.187 **	0.300 **
Peer	0.164 **	0.150 **
Prosocial	0.131 **	0.124 **
Positive Well-Being	−0.196 **	−0.282 **
Negative Well-Being	0.178 **	0.299 **
Physical Health	−0.166 **	−0.255 **
Flourishing	−0.227 **	−0.289 **

**. Correlation is significant at the 0.01 level (2-tailed).

**Table 3 brainsci-15-00609-t003:** Correlations between anxiety, depression, AQ, and ADHD and the predictors of well-being.

	Anxiety	Depression	Total AQ	Total ADHD
Anxiety	1	0.593 **	0.181 **	0.296 **
Depression	0.593 **	1	163 **	312 **
Total AQ	0.181 **	0.163 **	1	0.310 **
Total ADHD	0.296 **	0.312 **	0.310 **	1
Stress	0.538 **	0.525 **	0.150 **	0.315 **
Social Support	−0.096 *	−0.162 **	−0.149 **	−0.217 **
Positive Coping	−0.239 **	−0.274 **	−0.203 **	−0.245 **
Negative Coping	0.503 **	0.424 **	0.148 **	0.216 **
Psychological Capital	−0.419 **	−0.473 **	−0.232 **	−0.273 **
Work–Life Balance	0.263 **	0.283 **	0.068	0.245 **
Workload	0.380 **	0.390 **	0.201 **	0.332 **
Sleepiness	0.396 **	0.324 **	0.141 **	0.304 **
Flow	−0.133 **	−0.188 **	−0.149 **	−0.236 **
Rumination	−0.132 **	−0.160 **	−0.073	−0.120 **

*. Correlation is significant at the 0.05 level (2-tailed). **. Correlation is significant at the 0.01 level (2-tailed).

**Table 4 brainsci-15-00609-t004:** Regression with AQ, ADHD, anxiety, and depression and the predictor variables and outcomes.

Outcome	Predictor	Beta	*p*-Value	VIF
Positive Well-Being	Anxiety Depression	0.173 0.406	<0.001 <0.001	1.552 1.563
Negative Well-Being	Anxiety Depression	0.287 0.371	<0.001 <0.001	1.551 1.564
Physical Health	Depression ADHD	0.242 0.120	<0.001 0.007	1.563 1.216
Flourishing	Depression AQ	0.452 0.112	<0.001 0.003	1.571 1.121
Hyperactivity	Depression AQ ADHD	0.135 0.177 0.406	0.003 <0.001 <0.001	1.572 1.119 1.222
Conduct	Depression ADHD	0.222 0.267	<0.001 <0.001	1.565 1.216
Emotional	Anxiety Depression	0.475 0.345	<0.001 <0.001	1.547 1.564
Peer Problems	Depression AQ	0.229 0.116	<0.001 0.009	1.576 1.122
Prosocial Behaviour	Anxiety AQ	0.175 0.113	<0.001 0.013	1.552 1.121

**Table 5 brainsci-15-00609-t005:** Regression with AQ, ADHD, anxiety, and depression and established predictors.

Outcome	Predictor	Beta	*p*-Value	VIF
Stress	ADHD	0.121	0.002	1.217
Social Support	ADHD	0.155	<0.001	1.223
Positive Coping	AQ ADHD	0.133 0.141	0.003 0.002	1.120 1.217
Work–Life Balance	ADHD	0.190	<0.001	1.215
Workload	ADHD	0.200	<0.001	1.213
Sleepiness	ADHD	0.187	<0.001	1.211
Flow	ADHD	0.163	<0.001	1.207

**Table 6 brainsci-15-00609-t006:** Regression with AQ, ADHD, anxiety, depression, and established WPQ predictors.

Outcome	Predictors	Beta	*p*-Value
Positive Well-Being	Depression Social Support Psychological Capital	−0.280 0.104 0.219	<0.001 0.009 <0.001
Negative Well-Being	Depression Stress Psychological Capital Sleepiness	0.233 0.229 −0.125 0.106	<0.001 <0.001 0.005 0.005
Physical Health	Psychological Capital	0.216	<0.001
Flourishing	Depression Psychological Capital Flow Rumination	−0.269 0.327 0.192 0.090	<0.001 <0.001 <0.001 0.007
Hyperactivity	AQ ADHD Workload	0.154 0.328 0.184	<0.001 <0.001 <0.001
Conduct Problems	ADHD	0.194	<0.001
Emotional Problems	Anxiety Depression Stress Negative Coping Sleepiness	0.319 0.254 0.110 0.155 0.097	<0.001 <0.001 <0.001 <0.001 <0.001
Peer Problems	Stress Psychological Capital	−0.195 0.214	<0.001 <0.001
Prosocial Behaviour	Positive Coping Flow	0.156 0.187	0.002 <0.001

## Data Availability

The data presented in this study are available on request from the corresponding author. The data are not publicly available due to privacy guidelines, as the data was collected through the university.

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
