# Peer review of "Mental Health Correlates of Autistic and ADHD Traits in Secondary School Students"

_brainsci, 2025, doi:10.3390/brainsci15060609_

Round 1

Reviewer 1 Report

Comments and Suggestions for Authors

This study illustrates a significant correlation between anxiety, depression, AQ and ADHD through a survey of 560 secondary school students. Anxiety and depression were also significantly correlated with all well-being and SDQ variables. This study has some research value as a reference. However, there are many issues that need to be addressed in this paper:

1. Although the Autism Quotient was administered to 560 secondary school students in this study, there were no individuals with a diagnosis of autism among the 560 secondary school students, so are these test results representative of the data for individuals with autism?

2. What is the purpose of parts II-VII in the Introduction section? It feels like an review article. Suggest deleting or streamlining to introduction.

3. The results section of this paper uses a lot of tables to present statistical results, and it is recommended that bar charts be drawn to present the variability between groups.

4. The summarizing statement in Table 7 in the discussion section is too cumbersome and it is suggested that it be condensed and replaced by a symbol.

5. The format of references in this paper is also inconsistent, with some references having doi and others not. Please standardize the format of references.

Reviewer 2 Report

Comments and Suggestions for Authors
  1. The Introduction is overly lengthy and contains redundant information, which affects the clarity and logical flow.
  2. Please report the internal consistency (Cronbach's alpha) of each questionnaire used in the current sample.
  3. Please clarify the appropriateness of each questionnaire for use with adolescents. Specifically, using the Adult ADHD Self-Report Scale (ASRS) to assess ADHD symptoms in adolescents may not be suitable and requires justification or an alternative measure.
  4. Please clarify the scoring methods for the AQ and ADHD questionnaires. The AQ is a 10-item scale with 4-point ratings—why is the total score reported as ranging from 0 to 10? For the ADHD scale, it appears that only 6 items were used—please provide the rationale and supporting evidence for this, and specify the total score range.
  5. Please clarify the rationale and provide evidence supporting the adequacy of using only one item per sub-scale of the SDQ, which originally consists of five items per sub-scale. What was the basis for this adaptation, and how was the reliability and validity of this shortened version evaluated?
  6. Please specify the total number of items in the WPQ and provide details on how many items were included in each of the 16 sub-scales (i.e., anxiety, depression, the 10 predictors, and the 4 outcomes).
  7. The current study measured anxiety and depression using only one item each from the WPQ, which raises concerns about the reliability and validity of these assessments. In contrast, other constructs such as ADHD and autistic traits were measured using standardized multi-item scales. Given the importance of anxiety and depression in the theoretical framework, it is recommended that the authors use well-validated multi-item instruments (such as the Beck Depression Inventory (BDI) and Beck Anxiety Inventory (BAI)), or provide a clear rationale and evidence supporting the adequacy of the single-item measures.
  8. The study conducted multiple correlation and regression analyses, but no correction for multiple comparisons was reported.
  9. It would be beneficial to include a mediation analysis using Structural Equation Modeling (e.g., using AMOS software), with predictors as independent variables, outcomes as dependent variables, and AQ, ADHD, anxiety, and depression as mediating variables, to systematically clarify the relationships among these variables.
  10. What tools were used for measuring well-being and mental health in the current study? Or are they interchangeable terms, measured by the WPQ? What is the purpose and significance of measuring SDQ in this study?

Reviewer 3 Report

Comments and Suggestions for Authors

Dear Authors,

Thank you for the opportunity to review your manuscript. Your study provides important insights into the relationship between neurodevelopmental traits and mental health outcomes in secondary school students. The research design and analysis are appropriate, and the topic is both timely and significant. However, several areas require revision to improve the clarity, rigor, and overall presentation of the paper.

  • Abstract: Please revise the abstract to concisely summarize key findings, such as the stronger associations of anxiety and depression compared to AQ and ADHD traits. Including brief methodological details, such as sample size and measures used, would strengthen the abstract.
  • Introduction: The introduction offers a comprehensive background, but it would benefit from a more focused narrative. Consider condensing sections that describe historical aspects of autism and emphasize more clearly how your study fills a specific research gap.
  • Participants and Ethics: Please specify the exact age range of the participants. Additionally, clarify whether parental or guardian consent was obtained, which is crucial when working with minors.
  • Measures: Although validated instruments were used, it would enhance the methodological rigor to report internal consistency measures (e.g., Cronbach’s alpha) or cite previous validation studies.
  • Statistical Analysis: Given the inclusion of highly correlated variables (AQ, ADHD, anxiety, depression), please state whether multicollinearity diagnostics (e.g., VIFs) were performed prior to regression analysis.
  • Results: Address inconsistencies in sample sizes reported across tables. It is also recommended to explain the approach to handling missing data. In reporting regression results, consider providing additional information such as R² and model fit indices where possible. Language implying causality (e.g., "predicted") should be rephrased to reflect associative findings.
  • Discussion: Frame the interpretation of results with greater caution, avoiding causal implications. Expand the discussion on practical implications, specifically how these findings might guide mental health interventions for adolescents exhibiting neurodevelopmental traits. It would also be useful to acknowledge regional limitations (e.g., sample from South Wales) and suggest specific avenues for future research, such as longitudinal and mediation models.
  • Limitations: Although limitations are discussed, emphasize more clearly the cross-sectional design's restriction on causal inference, and address the inherent biases of self-reported measures in adolescent samples.
  • References: Ensure consistency in the formatting of references, particularly regarding DOI links and the presentation style required by the journal.
  • Language and Presentation: The manuscript would benefit significantly from careful English editing. Improving sentence flow, removing redundancy, and enhancing clarity will make the manuscript more readable and professionally polished.

Overall, your study is valuable and has the potential to make an important contribution to the literature. I encourage you to address the issues outlined above to strengthen the manuscript for potential publication.

Round 2

Reviewer 1 Report

Comments and Suggestions for Authors

 Accept 

Author Response

Thank You

Reviewer 3 Report

Comments and Suggestions for Authors

Thank you to the authors for their thoughtful responses and substantial revisions. The manuscript has improved in many key areas, including overall structure, ethical transparency, and result presentation. Several of the initial concerns have been addressed. However, a few important points remain, which should be resolved in a final round of revision prior to acceptance.

1. Abstract

The abstract does not yet fully highlight the main comparative finding: that anxiety and depression traits were more strongly associated with well-being outcomes than autistic or ADHD traits. This is a central contribution of the study and should be explicitly stated.

Additionally, the abstract uses multiple abbreviations—AQ, ADHD, SDQ, WPQ—without providing full names on first mention. Since Brain Sciences is a multidisciplinary journal read by researchers across neuroscience, psychology, and behavioral sciences, these abbreviations may not be immediately recognizable to all readers.

Moreover, the abstract would benefit from a brief statement of the study’s contribution to the literature—for example, how this research extends prior findings in university populations by focusing on a younger, school-based sample, or how it informs early intervention approaches in adolescents with subclinical traits.

Recommendation:

  • Clearly state the key comparative finding.
  • Spell out all abbreviations on first mention (e.g., attention-deficit/hyperactivity disorder (ADHD), Autism Spectrum Quotient (AQ), Strengths and Difficulties Questionnaire (SDQ), Well-being Process Questionnaire (WPQ)).
  • Briefly articulate the study’s contribution to the literature or practical relevance.

2. Participant Description – Clinical vs. General Population

It remains somewhat unclear whether the participants included any individuals with formal clinical diagnoses or were drawn entirely from the general secondary school population. Given that screening tools (AQ, ASRS) were used rather than diagnostic interviews, this distinction should be clearly stated.

Recommendation:

  •  Clarify that the sample consisted of general population students
  •  Indicate that traits were assessed using self-report screening tools, not clinical diagnoses

3. Rationale for Focusing on Secondary School Students

Although previous studies cited in the introduction include university students, the rationale for focusing on secondary school students specifically is not sufficiently developed. Adolescence is a key developmental stage for the onset of mental health issues, but this context is only implicitly referenced.

Recommendation:

  • Explain why secondary school students are a theoretically meaningful population
  • Reference adolescence as a sensitive developmental period for mental health

4. Multicollinearity

The authors now note that no multicollinearity analyses (e.g., VIFs) were conducted, and this is acknowledged in the manuscript. However, no rationale is provided for this omission.

Recommendation:

  •  Add a brief justification for not conducting multicollinearity diagnostics

5. Internal Consistency / Reliability

Although the manuscript includes a clearer description of the screening instruments used (e.g., AQ-10, ASRS, SDQ), it does not report internal consistency metrics such as Cronbach’s alpha. The authors argue that reliability is not relevant for instruments where individual items were analyzed (e.g., WPQ, SDQ sub-items), which is understandable. However, for tools where total or subscale scores are used in regression analyses—as is the case for the AQ-10, ASRS, and SDQ—reporting internal consistency is considered standard practice in psychometric research.

This is especially important because:

The reliability of scales can vary depending on sample characteristics (e.g., age, language, cultural context).

Screening instruments like the AQ and ASRS are often used with summed scores in non-clinical populations, and reporting reliability values helps readers assess the validity of these measures in the current sample.

Even if α values from prior studies are cited, it would strengthen the manuscript to either report Cronbach’s α for this study’s sample or at least discuss it as a limitation if not assessed.

Recommendation:

  • Please provide internal consistency estimates (e.g., Cronbach’s alpha) for all multi-item scales used in the regression analyses (AQ-10, ASRS, SDQ), or cite previous validation studies reporting reliability in adolescent or general population samples. If such data were not available, a rationale and a note in the limitations section would be appropriate.

I look forward to seeing the final, polished version of this valuable contribution to adolescent mental health research.
